# Intraoperative Transbronchial Metallic Coil Marking for Small Peripheral Pulmonary Lesions in a Hybrid Operation Room

**DOI:** 10.3390/cancers16234038

**Published:** 2024-12-01

**Authors:** Naoya Kawakita, Hiroaki Toba, Naoki Miyamoto, Shinichi Sakamoto, Hiroyuki Sumitomo, Taihei Takeuchi, Atsushi Morishita, Ayaka Baba, Emi Takehara, Keisuke Fujimoto, Masakazu Goto, Hiromitsu Takizawa

**Affiliations:** 1Department of Thoracic and Endocrine Surgery and Oncology, Institute of Biomedical Sciences, Tokushima University Graduate School, Kuramoto-cho, Tokushima 770-8503, Japan; kawakita.naoya@tokushima-u.ac.jp (N.K.);; 2Department of Oncological Medical Services, Institute of Biomedical Sciences, Tokushima University Graduate School, Kuramoto-cho, Tokushima 770-8503, Japan

**Keywords:** bronchoscopy, metallic coil, cone-beam computed tomography, small peripheral pulmonary lesion, image-guided video assisted thoracic surgery, hybrid operation room

## Abstract

The present study examined the efficacy of a transbronchial metallic coil marking method for identifying small peripheral pulmonary lesions using cone–beam CT (CBCT) in a hybrid operating room. This technique enables accurate localization and resection of deeply located lesions. The procedure involved inserting an ultrathin bronchoscope under virtual navigation, placing the coil via CBCT, and performing wedge resection through video-assisted thoracoscopic surgery under fluorescence guidance. In total, 87 patients with 90 lesions, primarily small (median size 11 mm), were included. All lesions were successfully visualized and marked, with a median distance from lesions to coils of 3.6 mm and a marking time of 23.5 min. No complications arose during the marking procedure, demonstrating that CBCT is a viable alternative for identifying peripheral lung lesions while ensuring minimally invasive treatment under general anesthesia.

## 1. Introduction

With improvements in the performance of computed tomography (CT) scanners, smaller peripheral lung lesions are now detectable. In addition, low-dose CT screening has recently contributed to reductions in lung cancer deaths [1,2]. This social background suggests that the number of small peripheral lung lesions requiring a pathological diagnosis will continue to increase. It remains challenging to identify small pulmonary lesions intraoperatively and perform precise resection. Various marking methods have been applied to cases in which it is difficult to identify the location of a lesion by a visual inspection or palpation during surgery [3].

We previously reported a method involving transbronchial metallic coil marking followed by video-assisted thoracoscopic surgery (VATS) resection [4,5]. This method is useful for identifying deeply located lesions and ensuring sufficient margins. Its advantages include the ability to select resection margins in segmentectomy and wedge resection for multiple lesions, its safety, and the low risk of complications. However, previous methods required patients to undergo the bronchoscopic marking procedure under local anesthesia in a CT-equipped interventional radiology suite several days before surgery [4,5].

Minimally invasive surgery has recently been performed under image guidance in a hybrid operating room (HOR). Imaging in an HOR is also applied to the intraoperative identification of small peripheral lung lesions [6,7,8,9]. Since October 2016, our hospital has been equipped with an HOR that utilizes cone–beam CT (CBCT), allowing us to perform transbronchial metallic coil marking in a one-stop manner. Therefore, patients may now undergo a full range of procedures, from marking to lung resection, under general anesthesia.

The present study investigated the results of transbronchial metallic coil marking using imaging techniques including CBCT in an HOR.

## 2. Materials and Methods

### 2.1. Eligibility

Indications for transbronchial metallic coil marking include (i) peripheral pulmonary lesions measuring ≤10 mm that are scheduled for VATS wedge resection, (ii) peripheral pulmonary nodules located up to 30 mm from the pleural surface, and (iii) ground-glass nodules (GGNs) without pleural changes. Segmentectomy was adopted for lesions located relatively deep and where securing margins was expected to be difficult. This method may not be applicable to patients whose bronchi do not reach the vicinity of the target lesion because of the difficulties associated with placing the coil at the appropriate site. Therefore, the relationship between bronchi and the target lesion needs to be carefully confirmed on preoperative multi-detector CT (MDCT) images. However, during the study period, few cases were excluded for the aforementioned reason. This clinical trial was approved by the Committee for Medical Ethics of Tokushima University Hospital (No. 3672). After obtaining informed consent, 87 patients underwent transbronchial metallic coil marking for 90 lesions between October 2016 and December 2022.

### 2.2. Patients

Patients consisted of 39 males and 48 females, with a median age of 71 (interquartile range [IQR] 63–76) years. No lesions were histologically diagnosed preoperatively. Eighty-four patients had solitary lesions, and three had 2 lesions. The median tumor size was 11 (IQR: 8–15) mm. The median distance from the tumor to the pleural surface was 8.7 (IQR: 4–15) mm. A total of 29 lesions were located in the right upper lobe, 4 in the right middle lobe, 18 in the right lower lobe, 22 in the left upper lobe, and 17 in the left lower lobe. Lesions consisted of 19 pure GGN, 35 partly solid, and 36 solid types (Table 1).

### 2.3. Virtual Bronchoscopic Navigation

VBN systems were used to lead the tip of the bronchoscope to the nearest bronchus from the target lesion. Chest CT was performed with an MDCT scan (Aquilion; Toshiba Medical Systems, Tokyo, Japan) with the following parameters: collimation of 0.5 mm, 4 detectors, pitch of 5 to 7, and a rotation time of 0.5 s. Data were converted to DICOM data and VBN images were reconstructed with Bf-NAVI software;https://www.olympus.co.jp/jp/news/2008a/nr080612bfnavij.html; accessed on 1 November 2024 (Olympus Medical Systems, Tokyo, Japan) or SYNAPSE VINCENT bronchoscopic navigation-dedicated software; https://www.fujifilm.com/jp/ja/healthcare/healthcare-it/it-3d/vincent; accessed on 1 November 2024 (Fujifilm Medical, Tokyo, Japan).

### 2.4. Bronchoscopic Metallic Coil Marking Procedure

The bronchoscopic marking procedure was performed in an HOR (Figure 1) consisting of an X-ray angiography imaging system and surgical table (Alpheid Hybrid+, INFX-8000H; Canon Medical Systems Corporation, Tochigi, Japan). This system enabled CBCT imaging. Patients were intubated with a single-lumen tube in the supine position, and general anesthesia was induced. Because the movement of the bronchoscope is sometimes restricted through a double-lumen tube and it cannot reach the target bronchus, we prefer to use a single-lumen tube during the marking procedures. The patient’s upper limbs were positioned along the trunk, and the C-arm was carefully adjusted to avoid contact with the patient’s body during C-arm rotation. The first CBCT scan was taken by rotating the C-arm to confirm the visibility of the lesion. To obtain clear CBCT images, airway pressure was maintained at 20 mmHg during C-arm rotation. CBCT scans were performed with a 5 s acquisition protocol. Cross-sectional images in 3 planes (vertical, frontal, and sagittal) were reconstructed automatically on a dedicated workstation, and images were displayed on a monitor in front of the examiners.

After an ultrathin video-bronchoscope (type XP260F; Olympus, Tokyo, Japan) was inserted into the objective bronchus guided by VBN, a coil-feeding microcatheter (MST125/27HFR; Asahi Intecc, Nagoya, Japan) was inserted through the bronchoscope’s working channel, and the tip of the microcatheter was adjusted using C-arm fluoroscopy guidance. A second CBCT scan was then taken to confirm the position of the catheter tip. The surgeon held the bronchoscope and catheter in place only during the second CBCT scanning to confirm the position of the catheter tip. If the catheter tip was not close to the lesion, the tip position was adjusted using CBCT images as a reference, and an additional CBCT scan was taken if necessary.

After confirming that the tip of the microcatheter reached the lesion on multi-planar reconstructions of CBCT (axial, coronal, and sagittal), a fibered platinum coil (Boston Scientific, Tokyo, Japan) was positioned under fluoroscopic guidance (Figure 2A). A third CBCT image was acquired to confirm the location of the deployed metallic coil relative to the lesion (Figure 2B,C). A 3D reconstruction of post-marking CBCT images of the lesion and coil was displayed on a monitor in front of the surgeons (Figure 2D).

### 2.5. VATS Resection Technique

The patient was re-intubated with a double lumen tube and placed in the left lateral decubitus position. Pulmonary wedge resection was performed using single- or 3-port VATS. Through fluoroscopic 2D imaging on the monitor, the location of the metallic coil was identified. The nodule with the metallic coil was grasped with pulmonary forceps and resected with endostaplers under fluoroscopic guidance (Figure 3A,B). Lung tissue at which the coil was located was manipulated with the forceps as few times as possible to reduce the risk of coil migration. The resected specimen was confirmed by fluoroscopy to contain the coil (Figure 3C) and then cut to confirm the presence of the lesion and the tumor-to-resection margin distance (Figure 3D). In our hospital, we aim to secure a 2 cm margin during resection, and have determined a margin of at least 1 cm or greater than the tumor diameter to be sufficient on a cut surface of the specimen [10,11,12,13]. In cases of suspected primary lung cancer requiring conversion to lobectomy, intraoperative rapid diagnosis was performed. However, in the majority of cases, final pathologic evaluation was performed on formalin-fixed, paraffin-embedded tissue sections.

### 2.6. Statistical Analysis

Descriptive statistics were used to provide a summary of the study variables. Continuous variables are expressed as medians and IQRs, whereas categorical data are given as counts and percentages.

## 3. Results

Figure 4 shows preoperative MDCT and CBCT findings. All 90 lesions were identified by CBCT, even small pure GGN lesions. Table 2 shows the results of coil marking, the surgical procedure, and histology. We successfully instilled coils into the objective bronchus in all lesions. The number of instilled coils per lesion was one in 84 cases and two in 3 cases. The median distance from the lesion to the coil was 3.6 (IQR: 0–8.2) mm. The median time required for the marking procedure was 23.5 (IQR: 18–37) min. There were no complications associated with the marking procedure.

All lesions were definitively identified, and wedge resection was performed with sufficient margins in all cases during thoracoscopic surgery. The median time for partial resection was 38 (IQR: 28–49) min.

Regarding the results of the intra-operative histological diagnosis, two cases with solid lesions diagnosed as primary lung cancer were converted to lobectomy. In recent years, it has been reported that segmentectomy or wedge resection can be expected to have the same prognosis as lobectomy for primary lung cancers smaller than 2 cm, even if the lesion is solid [14,15]; however, lobectomy was the standard for patients during the period included in this study. Lesions were histologically diagnosed as follows: 57 as primary lung cancer, 26 as metastatic tumors, 3 as nodular lymphoid hyperplasia, and 4 as others. There was no locoregional recurrence (median follow-up time: 47.2 months).

## 4. Discussion

Metallic coil marking under bronchoscopy is a useful method for identifying small peripheral lung lesions, even if it is changed to a one-stop method using an HOR. Patients may now undergo the entire procedure, from marking to resection of the lesion, under general anesthesia, thereby avoiding the discomfort of bronchoscopic procedures under local anesthesia. Using our previous methods, in a CT-equipped interventional radiology suite, bronchoscope marking was conducted under CT guidance, and the patient had to be moved into the gantry with the examination table to take a CT image after the bronchoscope and microcatheter reached the target bronchus [4,5]. In contrast, CBCT-guided procedures from the insertion of the bronchoscope to metallic coil marking are performed without moving the examination table or the position of the operator or assistant because the fluoroscopic device and CBCT imaging device are the same. This seamless procedure reduces the likelihood of accidents, such as dislocating the catheter tip, allowing for a more stable procedure.

CBCT images display CT images in three directions immediately after capture, which is useful for understanding the positional relationship between the lesion and the marker in three dimensions. CBCT has been used to mark small peripheral lung lesions, including dye injection via a percutaneous approach and dye or radiofrequency identification placement via a transbronchial approach [16,17]. Yang et al. reported localization of small pulmonary nodules by CBCT-guided bronchoscopy under general anesthesia followed by VATS with HOR. In their study, microcoils were placed at the deep margins in 16 patients [18], which was similar to our method. The resolution of CBCT is lower than that of MDCT; however, in the present study, even small GGNs were identified in the first CBCT images obtained. The dorsal lung is poorly aerated due to the effects of gravity and, thus, difficulties were associated with visualizing lesions in that area. To avoid this, airway pressure was kept at 20 mmHg during CBCT imaging to maintain aeration of the peripheral lungs as much as possible. Although artifacts from the bronchoscope, microcatheter, or coils sometimes increased the difficulty of visualizing lesions, comparisons with the preoperative MDCT provided information on the positional relationship between lung lesions and each device in all cases. Zhao et al. reported that CBCT in hybrid surgery was useful for the precise localization of small lung lesions [19]. Fang et al. also demonstrated the utility of CBCT in locating GGNs and their subsequent removal by VATS [20].

The transbronchial approach has the advantage of safety. No marking-related complications were encountered in the present study. Furthermore, the transbronchial approach has a lower risk of complications, such as pneumothorax and air embolisms, than the percutaneous approach [4,5,9]. Pneumothorax occurs in between 9.5 and 70% of cases with the percutaneous approach [21,22,23]. Therefore, the transtracheal approach is safer, particularly when the background lung is accompanied by emphysematous changes. Another complication of the percutaneous approach is pulmonary parenchymal hemorrhage (16.1% to 35%). Air embolisms are a rare and potentially fatal complication [24]. In addition, pleural seeding and needle placement may occur as adverse events with the percutaneous approach.

In the transbronchial approach, depending on the location of the lesion and its position relative to the bronchus, the tip of the bronchoscope may not reach the target bronchus. However, our method uses an ultra-thin bronchoscope with an outer diameter of 2.8 mm, which is guided by VBN, making it possible to selectively guide the tip of the bronchoscope up to the seventh branch. Therefore, the distance between the coil and lesion was very close, at a median of 3.6 mm, which is an important point that facilitates the precise resection of small lesions.

The metallic coil allows for accurate and pinpoint localization, unlike liquid dyes, which may spread to the surrounding lung parenchyma, particularly when used in emphysematous lungs. If the dye is injected away from the pleura, it may not be visible, and conversely, if the dye is injected just below the pleura and leaks into the pleural cavity, it may obscure the marking site [25,26].

One of the disadvantages of this method is radiation exposure during CBCT imaging. Although radiation exposure during CBCT was previously shown to be similar to that of low-dose CT [27], the number of CBCT scans during this procedure needs to be minimized in order to reduce the exposure of patients and surgeons to radiation. The use of bronchoscope holders and x-ray radiation shielding screens also needs to be considered in order to reduce the exposure of surgeons to radiation. Another disadvantage is the high cost of the metallic coils and microcatheters (approximately USD 500).

The present study has some limitations that need to be addressed. This was a single-center, retrospective study. Furthermore, our hospital has had long-term experience performing this procedure in a CT-equipped interventional radiology suite, which may have contributed to the favorable results obtained. In addition, since this study did not have a control group, the superiority of this method over other methods cannot be demonstrated. In this study, lymph node evaluation was not performed except for two lobectomy cases. Solid lesions of 1 cm or more have a high frequency of lymph node metastasis, and lymph node dissection is considered necessary [28]. On the other hand, for lesions with predominantly GGO, the frequency of lymph node metastasis is low and wedge resection is acceptable [29]. FDG-PET/CT is also useful for imaging diagnosis of lymph node metastasis [30]. In this study, most patients presented with predominant ground-glass opacities (GGO), and many solid lesions were metastatic pulmonary tumors. Additionally, as the study targeted cN0 cases based on imaging modalities such as FDG-PET/CT, lymph node evaluations were not performed. However, no local recurrences were observed, suggesting that the results were acceptable.

## 5. Conclusions

CBCT represents an alternative modality for identifying peripheral lung lesions because it has the ability to visualize even small GGNs. This method is a minimally invasive technique because the treatment sequence is completed under general anesthesia with the same quality as previous methods performed in CT-equipped interventional radiology suites.

## Figures and Tables

**Figure 1 cancers-16-04038-f001:**
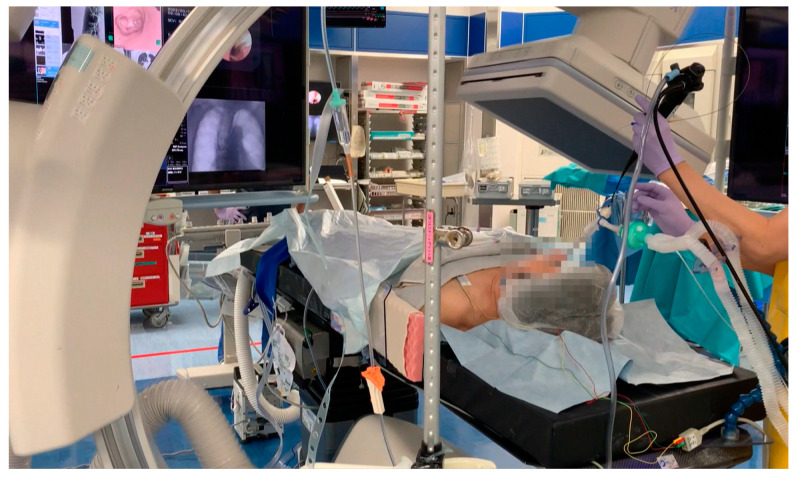
Bronchoscopic marking in a hybrid operation room.

**Figure 2 cancers-16-04038-f002:**
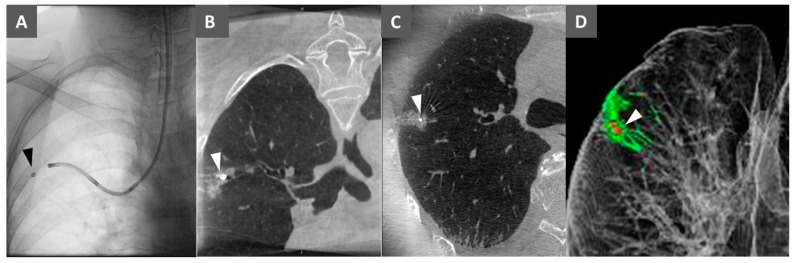
Transbronchial metallic coil marking. (**A**) A fluoroscopic image showing an ultrathin bronchoscope and microcoil (arrowheads). (**B**) The coronal plane of a cone–beam CT image showing a small pulmonary lesion and microcoil (arrowheads). (**C**) The axial plane of a cone–beam CT image showing a small pulmonary lesion and microcoil (arrowheads). (**D**) 3D reconstruction of post-marking CBCT images with the lesion and a microcoil (arrowheads).

**Figure 3 cancers-16-04038-f003:**
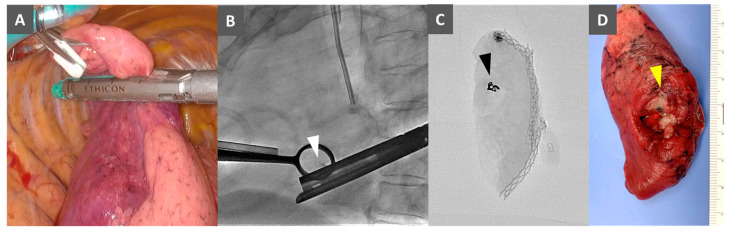
Images during video-assisted thoracoscopic surgery (VATS) pulmonary wedge resection. (**A**) A thoracoscopic image during VATS. (**B**) An intraoperative fluoroscopic image showing the metallic coil (arrow) grasped with pulmonary forceps. (**C**) A fluoroscopic image showing a microcoil (arrowhead) in a resected specimen. (**D**) A macroscopic image showing a microcoil (arrowhead) in a resected specimen.

**Figure 4 cancers-16-04038-f004:**
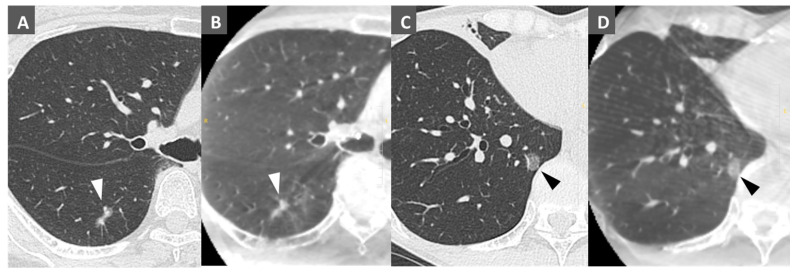
Comparison of multi-detector CT (MDCT) and cone–beam CT (CBCT) images. (**A**) MDCT and (**B**) CBCT of a small pulmonary lesion in right S6 (arrowheads). (**C**) MDCT and (**D**) CBCT of a small ground-glass nodule in right S10 (arrowheads).

**Table 1 cancers-16-04038-t001:** Characteristics of patients.

No. of patients (lesions)	87 (90)
Gender	
Male	39
Female	48
Age, median (IQR)	71 (63–76)
Lesion size, median (IQR) (mm)	11 (8–15)
Distance from lesion to pleura, median (IQR) (mm)	8.7 (4–15)
Location	
RUL/RML/RLL	29/4/18
LUL/LLL	22/17
Type of lesion	
Pure GGN	19
Part-solid GGN	35
Solid	36

RUL: Right upper lobe, RML: right middle lobe, RLL: right lower lobe, LUL: left upper lobe, LLL: left lower lobe, GGN ground-glass nodule.

**Table 2 cancers-16-04038-t002:** Results of the CBCT-CM, surgical procedure, and histology.

Marking procedure	
Instillation of coil into objective bronchi	90/90 (100%)
CBCT times, median (IQR)	3 (3–4)
Time of marking, median (IQR), (min)	23.5 (18–37)
Distance between lesion and microcoil, median (IQR) (mm)	3.6 (0–8.2)
Complication	0
Accurate identification of location	
Succeeded	90/90 (100%)
Time of partial resection, median (IQR) (min)	38 (28–49)
Histology	
Primary lung cancer	57
Metastatic tumor	26
Nodular lymphoid hyperplasia	3
Others (inflammation, organized pneumonia)	4

IQR: interquartile range.

## Data Availability

The data presented in this study are available in this article.

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
