# Peer review of "Intraoperative Transbronchial Metallic Coil Marking for Small Peripheral Pulmonary Lesions in a Hybrid Operation Room"

_cancers, 2024, doi:10.3390/cancers16234038_

Round 1
Reviewer 1 Report
Comments and Suggestions for Authors
The authors report the series of transbronchial microcoil localization in the hybrid OR. The manuscript is well-written with adequate methodology/figures.
There are several suggestions to improve the quality before formal publication.
1. The authors mentioned about the inapplicable for the vicinity of the lesion could not be reached through bronchi. How many cases during the study period were inapplicable and not enrolled for this methods?
2.During the CBCT scan, how you place the bronchoscope and catheter in position? Did you use any scope holder? If it is, please mention about the details and provide the figure if available.
3. The authors mentioned about lung collapse over dependent part. How many cases with this situation and how to manage that except keeping high pressure (20mmHg)? Were collapsed area re-expansion? If not, how to salvage the localization and image reconstruction? Did authors ever try lateral position to prevent it?
4. Page 2, Line 83 The term " FATS-CM" is not noted for the full name in the manuscript.
5. Introduction (Page 2, Line 57-58) emphasize microcoil localization for deep lesion as its advantage, however the average depth was 8.7mm, which is relatively superficial. How to manage deeper lesions? Would you choose segmentectomy directly?
6. The patient was reintubated with double lumen tube (DLT) after localization. Why not use bronchial blocker? Additionally, ultrathin bronchoscopy with 2.8mm OD could also pass the channel of DLT, is it possible to use DLT for bronchoscopy localization in the very beginning?
7. Please provide the scanning protocol of the CBCT
8. Please cite reference " Yang S-M, et al. Single-stage augmented fluoroscopic bronchoscopy localization and thoracoscopic resection of small pulmonary nodules in a hybrid operating room. Eur J Cardiothorac Surg 2023; doi:10.1093/ejcts/ezac541." Which might also be one of the first references report transbronchial microcoil placement in the hybrid OR, as you mentioned in page 6 line 194.
Author Response
Reply to Reviewer 1:
We wish to express our profound appreciation to the Reviewer for the insightful comments, which have helped us to significantly improve our paper.
Comment 1: The authors mentioned about the inapplicable for the vicinity of the lesion could not be reached through bronchi. How many cases during the study period were inapplicable and not enrolled for this method?
Reply 1: We appreciate the reviewer’s comment. In fact, very few cases were excluded during the study period because the bronchus did not reach the vicinity of the lesion.
We added the following sentence.
Changes in the text: However, during the study period, few cases were excluded for the aforementioned reason. (lines 82 - 83)
Comment 2: During the CBCT scan, how you place the bronchoscope and catheter in position? Did you use any scope holder? If it is, please mention about the details and provide the figure if available.
Reply 2: The surgeon held the bronchoscope and catheter during CBCT scanning to confirm the catheter tip position. We recognize that this is an area that needs improvement, and have therefore described it as a disadvantage of our method in the Discussion section. (lines 245 - 250)
We added the following sentence.
Changes in the text: The surgeon held the bronchoscope and catheter in place only during the second CBCT scanning to confirm the position of the catheter tip. (lines 127 - 128)
Comment 3: The authors mentioned about lung collapse over dependent part. How many cases with this situation and how to manage that except keeping high pressure (20mmHg)? Were collapsed area re-expansion? If not, how to salvage the localization and image reconstruction? Did authors ever try lateral position to prevent it?
Reply 3: We appreciate the reviewer’s comments. Early in the study, we encountered a case in which the lesion was difficult to visualize due to insufficient dorsal lung expansion, but this issue was resolved by performing CBCT scans while maintaining airway pressure at 20 mmHg. Therefore, we have not performed this marking technique in the lateral position.
Comment 4: Page 2, Line 83 The term " FATS-CM" is not noted for the full name in the manuscript.
Reply 4: We would like to thank the reviewer for the advice. As the reviewer pointed out, FATS-CM was not defined; it is an abbreviation that we have used in previous reports.
We have made the corrections.
Changes in the text: After obtaining informed consent, 87 patients underwent transbronchial metallic coil marking for 90 lesions between October 2016 and December 2022. (lines 85 - 86)
Comment 5: Introduction (Page 2, Line 57-58) emphasize microcoil localization for deep lesion as its advantage, however the average depth was 8.7mm, which is relatively superficial. How to manage deeper lesions? Would you choose segmentectomy directly?
Reply 5: As the reviewer stated, we think that segmentectomy should be adopted for deep lesions where it is difficult to secure margins.
We added the following sentence.
Changes in the text: Segmentectomy was adopted for lesions located relatively deep and where securing margins was expected to be difficult. (lines 77 - 78)
Comment 6: The patient was reintubated with double lumen tube (DLT) after localization. Why not use bronchial blocker? Additionally, ultrathin bronchoscopy with 2.8mm OD could also pass the channel of DLT, is it possible to use DLT for bronchoscopy localization in the very beginning?
Reply 6: We appreciate the reviewer’s comments. Because the movement of the bronchoscope is sometimes restricted through DLT and it cannot reach the target bronchus, we prefer to use a single lumen tube during the marking procedures. We think the use of a bronchial blocker is an option as well. We think the use of a bronchial blocker is an option as well.
We added the following sentence.
Changes in the text: Because the movement of the bronchoscope is sometimes restricted through a double-lumen tube and it cannot reach the target bronchus, we prefer to use a single-lumen tube during the marking procedures. (lines 112 - 114)
Comment 7: Please provide the scanning protocol of the CBCT
Reply 7: We appreciate the reviewer’s comments. CBCT scans were performed with a 5-s acquisition protocol, the maximum speed of the instrument.
We added the following sentence.
Changes in the text: CBCT scans were performed with a 5-s acquisition protocol. (lines 118-119)
Comment 8: Please cite reference " Yang S-M, et al. Single-stage augmented fluoroscopic bronchoscopy localization and thoracoscopic resection of small pulmonary nodules in a hybrid operating room. Eur J Cardiothorac Surg 2023; doi:10.1093/ejcts/ezac541." Which might also be one of the first references report transbronchial microcoil placement in the hybrid OR, as you mentioned in page 6 line 194.
Reply 8: We appreciate the reviewer’s comments. As the reviewer commented, bronchoscopic coil marking in a hybrid operating room was reported by Yang et al. in 2022. We have deleted the statement that there have been no similar reports other than ours and changed it to the following statement.
Changes in the text: Yang et al reported localization of small pulmonary nodules by CBCT-guided bronchoscopy under general anesthesia followed by VATS with HOR. In this study, microcoils were placed at the deep margins in 16 patients [18], which is similar to our method. (lines 214 - 217)
- Yang, S.M.; Chung, W.Y.; Ko, H.J.; Chen, L.C.; Chang, L.K.; Chang, H.C.; Kuo, S.W.; Ho, M.C. Single-stage augmented fluoroscopic bronchoscopy localization and thoracoscopic resection of small pulmonary nodules in a hybrid operating room. Eur J Cardiothorac Surg. 2022. doi: 10.1093/ejcts/ezac541.
Thank you again for your comments on our paper. We hope that the revised manuscript is now suitable for publication.
Reviewer 2 Report
Comments and Suggestions for Authors
I agree with the authors on their statement in lines 194-195 on the originality of the proposed methods for marking small nodules using transbronchial coils. I couldn’t find comparable papers in a quick search in PubMed. I have a few comments on the manuscript, and I thank the authors for reading and considering them.
1. Your second indication for the technique is: “deeply located nodules between 0 and 30 mm from the pleural surface” (lines 75-76). I wouldn’t consider that distance as a deep location. Maybe you meant: “peripheral pulmonary nodules located up to 30mm from the pleural surface”?
2. In lines 143-144 I could read that “when necessary, lesions were submitted for a rapid diagnosis by frozen sections”. It would be advisable that you briefly specify when did you consider necessary a rapid pathological evaluation. I suppose that the reason for that is just to be sure that the specimen contains a sub-solid nodule.
3. Also, in lines 168-160 I could read that “wedge resection was performed with sufficient margins in all cases during thoracoscopic surgery”. What margin are you considering as sufficient for solid and sub-solid lesions? You could include a short comment on this topic in the Discussion section and this reference or a similar one, DOI: 10.1016/j.athoracsur.2017.04.024.
4. What was the reason to convert two cases to lobectomy (lines 171-172). According to current evidence, wedge and anatomical resection are considered equally effective for small-size N0 NSCLC. You also could include these references, or similar ones, in your discussion section: 10.1016/j.jtcvs.2023.07.008 and/or 10.1016/j.athoracsur.2024.03.008.
5. I could read nothing in your manuscript related to lymph node evaluation in lung cancer cases being the majority in your series. There is some debate in recently published papers on the topic. You could also discuss briefly the problem in the manuscript and include this reference or others DOI: 10.1016/j.athoracsur.2022.11.040.
6. In lines 207-210 you accurately comment on the advantages of transbronchial approach compared to the percutaneous one and include references 4 and 5. In your reference 9 the authors also compared the rate and type of complications for both approaches in their experience.
Author Response
Reply to Reviewer 2:
We wish to express our profound appreciation to the Reviewer for the valuable comments, which have helped us to significantly improve our paper.
Comment 1: Your second indication for the technique is: “deeply located nodules between 0 and 30 mm from the pleural surface” (lines 75-76). I wouldn’t consider that distance as a deep location. Maybe you meant: “peripheral pulmonary nodules located up to 30mm from the pleural surface”?
Reply 1: We agree with the Reviewer’s comments. We intended to mean "peripheral pulmonary nodules located up to 30mm from the pleural surface."
We have changed the text as follows.
Changes in the text: (ii) peripheral pulmonary nodules located up to 30mm from the pleural surface, (line 75-76)
Comment 2: In lines 143-144 I could read that “when necessary, lesions were submitted for a rapid diagnosis by frozen sections”. It would be advisable that you briefly specify when did you consider necessary a rapid pathological evaluation. I suppose that the reason for that is just to be sure that the specimen contains a sub-solid nodule.
Reply 2: In cases of suspected primary lung cancer requiring conversion to lobectomy, intraoperative rapid diagnosis was performed.
We have changed the text as follows.
Changes in the text: In cases of suspected primary lung cancer requiring conversion to lobectomy, intraoperative rapid diagnosis was performed. However, in the majority of cases, final pathologic evaluation was performed on formalin-fixed, paraffin-embedded tissue sections. (line 156-159)
Comment 3: Also, in lines 168-160 I could read that “wedge resection was performed with sufficient margins in all cases during thoracoscopic surgery”. What margin are you considering as sufficient for solid and sub-solid lesions? You could include a short comment on this topic in the Discussion section and this reference or a similar one, DOI: 10.1016/j.athoracsur.2017.04.024.
Reply 3: We appreciate the Reviewer’s comment. In accordance with the comment, the following sentence and references were added to the method section.
Changes in the text: In our hospital, we aim to secure a 2 cm margin during resection, and have determined a margin of at least 1 cm or greater than the tumor diameter to be sufficient on a cut surface of the specimen [10-13]. (lines 154 -156)
- Wolf, A.S.; Swanson, S.J.; Yip, R.; Liu, B.; Tarras, E.S.; Yankelevitz, D.F.; Henschke, C.I.; Taioli, E.; Flores, R.M.; I-ELCAP Investigators. The Impact of Margins on Outcomes After Wedge Resection for Stage I Non-Small Cell Lung Cancer. Ann Thorac Surg. 2017, 104, 1171-1178.
- Mohiuddin, K.; Haneuse, S.; Sofer, T.; Gill, R.; Jaklitsch, M.T.; Colson, Y.L.; Wee, J.; Bueno, R.; Mentzer, S.J.; Sugarbaker, D.J.; Swanson, S.J. Relationship between margin distance and local recurrence among patients undergoing wedge resection for small (2 cm) non-small cell lung cancer. J Thorac Cardiovasc Surg. 2014, 147, 1169-1175; discussion 1175-1177.
- Sawabata, N.; Maeda, H.; Matsumura, A.; Ohta, M.; Okumura, M.; Thoracic Surgery Study Group of Osaka University. Clinical implications of the margin cytology findings and margin/tumor size ratio in patients who underwent pulmonary excision for peripheral non-small cell lung cancer. Surg Today. 2012, 42, 238-244.
- El-Sherif, A.; Fernando, H.C.; Santos, R.; Pettiford, B.; Luketich, J.D.; Close, J.M.; Landreneau, R.J. Margin and local recurrence after sublobar resection of non-small cell lung cancer. Ann Surg Oncol. 2007, 14, 2400-2405.
Comment 4: What was the reason to convert two cases to lobectomy (lines 171-172). According to current evidence, wedge and anatomical resection are considered equally effective for small-size N0 NSCLC. You also could include these references, or similar ones, in your discussion section: 10.1016/j.jtcvs.2023.07.008 and/or 10.1016/j.athoracsur.2024.03.008.
Reply 4: We would like to thank the reviewer for the advice. The two cases of conversion to lobectomy were due to solid lesions in which a diagnosis of primary lung cancer was obtained by rapid pathology diagnosis. Of course, we are aware of the recent results of CALGB140503, but the patients in this study underwent surgery between 2016 and 2022, and the standard procedure at that time was lobectomy.
We have changed and added the text as follows.
Changes in the text: Regarding the results of the intra-operative histological diagnosis, 2 cases with solid lesions diagnosed as primary lung cancer were converted to lobectomy. In recent years, it has been reported that segmentectomy or wedge resection can be expected to have the same prognosis as lobectomy for primary lung cancers smaller than 2 cm, even if the lesion is solid [14,15], however; lobectomy was the standard for patients during the period included in this study. (line 185-190).
- Altorki, N.; Wang, X.; Damman, B.; Mentlick, J.; Landreneau, R.; Wigle, D.; Jones, D.R.; Conti, M.; Ashrafi, AS.; Liberman, M.; de Perrot, M.; Mitchell, J.D.; Keenan, R.; Bauer, T.; Miller, D.; Stinchcombe, TE. Lobectomy, segmentectomy, or wedge resection for peripheral clinical T1aN0 non-small cell lung cancer: A post hoc analysis of CALGB 140503 (Alliance). J Thorac Cardiovasc Surg. 2024, 167, 338-347.
- Saji, H.; Okada, M.; Tsuboi, M.; Nakajima, R.; Suzuki, K.; Aokage, K.; Aoki, T.; Okami, J.; Yoshino, I.; Ito, H.; Okumura, N.; Yamaguchi, M.; Ikeda, N.; Wakabayashi, M.; Nakamura, K.; Fukuda, H.; Nakamura, S.; Mitsudomi, T.; Watanabe, S.I.; Asamura, H.; West Japan Oncology Group and Japan Clinical Oncology Group. Segmentectomy versus lobectomy in small-sized peripheral non-small-cell lung cancer (JCOG0802/WJOG4607L): a multicentre, open-label, phase 3, randomised, controlled, non-inferiority trial. 2022, 399, 1607-1617.
Comment 5: I could read nothing in your manuscript related to lymph node evaluation in lung cancer cases being the majority in your series. There is some debate in recently published papers on the topic. You could also discuss briefly the problem in the manuscript and include this reference or others DOI: 10.1016/j.athoracsur.2022.11.040.
Reply 5: We appreciate the Reviewer’s insightful comments. In accordance with the comment, the following sentence was added as a limitation of this study to the discussion section.
Changes in the text: In this study, lymph node evaluation was not performed except for two lobectomy cases. Solid lesions of 1 cm or more have a high frequency of lymph node metastasis, and lymph node dissection is considered necessary [28]. On the other hand, for lesions with predominantly GGO, the frequency of lymph node metastasis is low and wedge resection is acceptable [29]. FDG-PET/CT is also useful for imaging diagnosis of lymph node metastasis [28]. In this study, most patients presented with predominant ground-glass opacities (GGO), and many solid lesions were metastatic pulmonary tumors. Additionally, as the study targeted cN0 cases based on imaging modalities such as FDG-PET/CT, lymph node evaluations were not performed. However, no local recurrences were observed, suggesting that the results were acceptable. (lines 262 - 271)
- Choi, S.; Yoon. D.W.; Shin, S.; Kim, H.K.; Choi, Y.S.; Kim, J.; Shim, Y.M.; Cho, J.H. Importance of Lymph Node Evaluation in 2-cm Pure-Solid Non-Small Cell Lung Cancer. Ann Thorac Surg. 2024, 117, 586-593.
- Suzuki, K.; Koike, T.; Asakawa, T.; Kusumoto, M.; Asamura, H.; Nagai, K.; Tada, H.; Mitsudomi, T.; Tsuboi, M.; Shibata, T.; Fukuda, H.; Kato, H.; Japan Lung Cancer Surgical Study Group (JCOG LCSSG). A prospective radiological study of thin-section computed tomography to predict pathological noninvasiveness in peripheral clinical IA lung cancer (Japan Clinical Oncology Group 0201). J Thorac Oncol. 2011, 6, 751-756.
- Schmidt-Hansen, M.; Baldwin, D.R.; Hasler, E.; Zamora, J.; Abraira, V.; Roqué, I, Figuls, M. PET-CT for assessing mediastinal lymph node involvement in patients with suspected resectable non-small cell lung cancer. Cochrane Database Syst Rev. 2014 doi: 10.1002/14651858.CD009519.pub2.
Comment 6: In lines 207-210 you accurately comment on the advantages of transbronchial approach compared to the percutaneous one and include references 4 and 5. In your reference 9 the authors also compared the rate and type of complications for both approaches in their experience.
Reply 6: We thank the reviewer for their comments. We have added reference 9 as a citation below.
Changes in the text: Furthermore, the transbronchial approach has a lower risk of complications, such as pneumothorax and air embolisms, than the percutaneous approach [4,5,9]. (lines 229 - 231)
Thank you again for your comments on our paper. We hope that the revised manuscript is now suitable for publication.